# A Comprehensive CNN Model for Age-Related Macular Degeneration Classification Using OCT: Integrating Inception Modules, SE Blocks, and ConvMixer

**DOI:** 10.3390/diagnostics14242836

**Published:** 2024-12-17

**Authors:** Elif Yusufoğlu, Hüseyin Fırat, Hüseyin Üzen, Salih Taha Alperen Özçelik, İpek Balıkçı Çiçek, Abdulkadir Şengür, Orhan Atila, Numan Halit Guldemir

**Affiliations:** 1Department of Ophthalmology, Elazig Fethi Sekin City Hospital, 23100 Elazig, Türkiye; elif.eraslan@yahoo.com; 2Department of Computer Engineering, Faculty of Engineering, Dicle University, 21000 Diyarbakır, Türkiye; huseyin.firat@dicle.edu.tr; 3Department of Computer Engineering, Faculty of Engineering, Bingol University, 12000 Bingol, Türkiye; huzen@bingol.edu.tr; 4Department of Electrical-Electronics Engineering, Faculty of Engineering, Bingol University, 12000 Bingol, Türkiye; sozcelik@bingol.edu.tr; 5Department of Biostatistics and Medical Informatics, Faculty of Medicine, Inonu University, 44000 Malatya, Türkiye; ipek.balikci@inonu.edu.tr; 6Department of Electrical-Electronics Engineering, Faculty of Technology, Firat University, 23100 Elazig, Türkiye; ksengur@firat.edu.tr; 7School of Electronics, Electrical Engineering and Computer Science, Queen’s University Belfast, Belfast BT9 5BN, UK; nguldemir01@qub.ac.uk

**Keywords:** ConvMixer, depthwise squeeze-and-excitation block, modified inception module, age-related macular degeneration, optical coherence tomography

## Abstract

**Background/Objectives**: Age-related macular degeneration (AMD) is a significant cause of vision loss in older adults, often progressing without early noticeable symptoms. Deep learning (DL) models, particularly convolutional neural networks (CNNs), demonstrate potential in accurately diagnosing and classifying AMD using medical imaging technologies like optical coherence to-mography (OCT) scans. This study introduces a novel CNN-based DL method for AMD diagnosis, aiming to enhance computational efficiency and classification accuracy. **Methods**: The proposed method (PM) combines modified Inception modules, Depthwise Squeeze-and-Excitation Blocks, and ConvMixer architecture. Its effectiveness was evaluated on two datasets: a private dataset with 2316 images and the public Noor dataset. Key performance metrics, including accuracy, precision, recall, and F1 score, were calculated to assess the method’s diagnostic performance. **Results**: On the private dataset, the PM achieved outstanding performance: 97.98% accuracy, 97.95% precision, 97.77% recall, and 97.86% F1 score. When tested on the public Noor dataset, the method reached 100% across all evaluation metrics, outperforming existing DL approaches. **Conclusions**: These results highlight the promising role of AI-based systems in AMD diagnosis, of-fering advanced feature extraction capabilities that can potentially enable early detection and in-tervention, ultimately improving patient care and outcomes. While the proposed model demon-strates promising performance on the datasets tested, the study is limited by the size and diversity of the datasets. Future work will focus on external clinical validation to address these limita-tions.

## 1. Introduction

The retina is a light-sensitive tissue lining the inner surface of the eye, responsible for capturing visual information. The macula, a specialized region of the retina, plays a key role in central vision, capturing high-resolution details such as colors and brightness levels, which are then transmitted to the brain for interpretation [1]. Age-related macular degeneration (AMD), a retinal condition, is characterized by degenerative changes occurring in the choroid and retina within the macular region [2]. As per the World Health Organization, roughly 1.3 billion people globally experience different levels of vision impairment, with AMD ranking as the third most significant contributor to visual loss [3,4]. This condition predominantly impacts individuals aged 50 and above, and its occurrence rises as people grow older [5]. With an aging population, AMD’s projected cases are estimated to reach 288 million by 2040, constituting approximately 8.7% of total global blindness cases [6,7].

AMD may be categorized into two forms: exudative, known as wet AMD, and non-exudative, known as dry AMD. Dry AMD is distinguished from wet AMD by the absence of blood or serum leakage. Around 85% to 90% of individuals with AMD have the dry form [8]. The presence of drusen in the retinal pigment epithelium layer is a characteristic feature of AMD, observed in both dry and wet forms. This drusen buildup can impair the function of the macula by disrupting its normal structure and causing progressive vision loss. While individuals with dry AMD might retain sufficient central vision, they often face significant limitations in functionality, including diminished night vision, fluctuations in sight, and challenges with reading due to a narrowed central vision area. Furthermore, a segment of dry AMD cases may advance to wet AMD as time progresses [9]. In individuals with wet AMD, the presence of blood or fluid leaks behind the macula can lead to the appearance of dark spots in their central vision. Choroidal neovascularization (CNV), which develops beneath the macula and retina, is one of the primary causes of wet AMD. This abnormal growth of blood vessels can result in macular edema and temporary vision impairment. Alternatively, it can lead to bleeding, posing a significant threat to the underlying photoreceptors and causing permanent vision loss. Wet AMD is known for causing rapid and escalating vision deterioration [10,11]. If CNV emerges in one eye, there is a heightened risk for the other eye, necessitating regular examinations. Consequently, consistent retinal screenings play a crucial role in detecting and treating AMD and in averting further deterioration [3].

Ophthalmologists now have access to various imaging technological advancements for detecting AMD-related features. Among the clinical-level imaging modalities, optical coherence tomography (OCT) and fundus retinal imaging (FRI) are the most commonly employed methods to identify AMD [12]. A limitation of fundus photography is that it provides a two-dimensional frontal view of the retina, whereas OCT offers higher resolution and sectional imaging, enabling detailed analysis of retinal layers. [13].

The OCT, an imaging technology widely employed in ophthalmology, serves to diagnose and track various retinal conditions in a noninvasive manner. This rapid medical imaging method utilizes low-coherence interferometry to produce detailed cross-sectional images of the retina and optic nerve head, covering the foremost segment of the visual pathway and evaluating visual disorders such as optic disc abnormalities both qualitatively and quantitatively [14]. Its capability to generate high-resolution 3D images offers precise insights into the structure and operation of the human retina [15]. The OCT technology generates 3D cross-sectional images of the eye’s organic tissues, enabling the scrutiny of various layers within the eye’s rear part. This capability facilitates the detection and ongoing observation of eye disorders and irregularities [16]. Especially in the case of AMD, OCT has emerged as the primary method for identifying and evaluating the condition, which can lead to vision problems and even blindness. Through precise imaging, OCT enables detailed visualization of retinal structures, facilitating early detection of abnormalities and aiding in the development of effective treatment plans [17,18]. This advanced technology has significantly improved diagnostic accuracy and supports ophthalmologists in providing targeted care for patients with retinal conditions [19].

As the prevalence of AMD increases, AI-based technologies have the potential to assist ophthalmologists by identifying subtle retinal features, thereby enhancing diagnostic efficiency. Concurrently, with the advancements in multimodal imaging and the emergence of new biomarkers, data volume per patient will expand [20]. As retinal scans become more detailed and widely utilized, the volume of OCT images to be processed is expected to increase, presenting challenges in efficiently diagnosing and treating AMD patients. Hence, there is a growing need for a computer-assisted diagnostic system capable of automatically detecting AMD from an extensive array of retinal OCT images [21].

The advancement of AI-based detection systems opens up new possibilities for AMD diagnosis and treatment [22]. AI-based detection techniques can address this need by facilitating early detection and intervention through scalable, affordable solutions for AMD diagnosis and monitoring [23]. CNN-based deep learning (DL) algorithms have demonstrated strong performance in detecting AMD and classifying its types (wet and dry) [24].

This study introduces a novel DL model utilizing CNN for AMD detection using OCT images. The model incorporates a combination of the Depthwise Squeeze-and-Excitation Block (DSEB), modified Inception module, and ConvMixer architecture. The key contributions of this proposed model (PM) are outlined as follows:An innovative method is presented, merging the benefits of reducing parameter count using depthwise separable convolutions with the multi-scale parallel design of the Inception module. This combined technique, known as the modified Inception module, effectively minimizes the number of trainable parameters, thus reducing computational costs.Incorporating a depthwise separable convolutions layer into the Squeeze-and-Excitation block within DSEB creates a sturdier configuration. The resultant DSEB architecture adeptly assigns significance to low-level features and integrates them with high-level ones, optimizing the utilization of low-level features while marginally affecting the computational load. Consequently, the suggested design bolsters both training efficacy and network efficiency. Empirical findings affirm that integrating this structure within the proposed architecture enhances classification accuracy.ConvMixer offers robust capabilities for extracting features, enabling the model to grasp intricate patterns and formations within the input images. This method excels in extracting features abundant in spatial details. Operating directly on input patches and employing standard convolutions solely for mixing steps, it significantly boosts classification accuracy. Moreover, the integration of depthwise separable convolution layers and a modified Inception module into the ConvMixer architecture curtails computational expenses, rendering the proposed architecture more straightforward and less computationally burdensome.Comprehensive experimental studies were conducted on a private AMD dataset consisting of 2316 images to analyze the classification performance of the PM. F1 score of 97.86%, recall of 97.77%, precision of 97.95%, and an accuracy of 97.98% were achieved with the PM. In addition, as a result of the experimental studies carried out on the public (Noor) AMD dataset, 100% results were obtained in all evaluation metrics. Comparisons with different DL models have shown the superior performance of the PM.

The structure of the remaining paper is laid out as follows: Section 2 presents an overview of related works extracted from the existing literature. Section 3 elaborates on the model introduced for AMD classification in this study, providing detailed insights into the DSEB, modified Inception module, and ConvMixer architecture incorporated into the PM. Section 4 encompasses our AMD dataset utilized in the experimental investigations and outlines the obtained experimental results. Lastly, Section 5 wraps up the article by presenting conclusions derived from the findings and suggesting potential paths for future exploration.

## 2. Related Works

DL has significantly developed medical image classification by enabling more accurate and efficient analysis of medical data. The continued development of DL models combined with advancements in medical imaging technologies holds significant promise for improving healthcare diagnostics and treatment. DL models rely on representation learning, wherein a neural network with multiple layers autonomously uncovers the necessary representations essential for classification, eliminating the need for manual crafting of features, thus replacing the multi-step process used in conventional methods [25]. DL, particularly CNN-based models, have demonstrated promising outcomes in the classification of AMD utilizing OCT images. There are CNN-based studies in the literature for the diagnosis and classification of AMD. A general summary of these studies is given in Table 1, Table 2, Table 3, Table 4 and Table 5.

## 3. Proposed Model (PM)

The PM consists of a combination of the ConvMixer (CM) architecture, modified Inception module (MIM), and Depthwise Squeeze-and-Excitation Block (DSEB), as shown in Figure 1 and Figure 2. DSEB consists of the combination of depthwise separable convolution (DSConv) and Squeeze-and-Excitation (SE) block. In the PM, the input image is taken as 224 × 224 × 3. Initially, a 2D Conv with a kernel size of 5 × 5 and 32 filters is applied to this input image. Subsequently, depthwise convolution (DConv) with a kernel size of 3 × 3, followed by a pointwise convolution (PConv) with a kernel size of 1 × 1 and 32 filters, Batch Normalization (BatchNorm), and a 3 × 3 kernel size for maximum pooling (max-pooling) are applied in sequence. The purpose of BatchNorm is to ease and accelerate the training process. The resulting feature map is then fed into an MIM comprising DSConv layers. The application of the MIM is illustrated in Figure 3. Within the scope of the PM, the MIM is applied once. Following this, a 2D Conv with a kernel size of 3 × 3 and 128 filters is applied to the obtained feature map. Next, it serves as input to the DSEB. Within the DSEB, two consecutive DSConv operations are applied. In the first DSConv, DConv with a kernel size of 5 × 5 and PConv with a kernel size of 1 × 1 and 64 filters are employed. In the second DSConv, DConv with a kernel size of 3 × 3 and PConv with a kernel size of 1 × 1 and 64 filters are applied. The output of the second DSConv is then fed into the SE block. Utilizing the SE block enhances the classification performance of networks with almost no increase in system cost. The structure of the DSEB is depicted in Figure 2. The feature map obtained at the output of the DSEB is given to the input of the CM. The creation of CM aimed to explore whether patches contribute to enhancing classification accuracy in image tasks [41]. Hence, the initial layer of CM serves as a patch embedding layer. In the initial phase of the CM approach, image patches are extracted via patch embedding. This embedding is executed utilizing a standard Conv layer with equivalent kernel size and stride as the patch size, mirroring the original CM. Subsequently, patch representation data are obtained. This is succeeded by employing a Gaussian Error Linear Unit (GELU) activation function and a BatchNorm layer. The model’s subsequent phase involves a CM layer block, comprising a residual block encompassing a DConv, a GELU, and a BatchNorm layer. The inputs are merged with the output from the BatchNorm layer. The combined output undergoes further processing via PConv, a GELU layer, and a BatchNorm layer. The CM block repetition occurs depth times. The final stage within PM is the classification phase. The feature map achieved at the conclusion of the second stage undergoes processing through Global Average Pooling (GAP) and fully connected (FC) layers. Ultimately, the output from the FC layer is subjected to a softmax layer, generating the classification prediction output. The hyperparameters of the CM layer in the PM are 5 for the kernel size of the DSConv and 256 for the number of filters of the PConv. Finally, the depth (d) of the CM layer is 16. In the first stage of the proposed CM, patch embedding, the image is divided into patches. The patch size (p) used in this block is 2. Moreover, comprehensive explanations of the techniques employed in the PM are outlined in subsequent sections.

### 3.1. Depthwise Squeeze-and-Excitation Block (DSEB)

The DSEB is an extension and refinement of the traditional SE block, often employed in CNN architectures to enhance feature representation. In contrast to the SE block, which predominantly operates across channels within a convolutional layer, the DSEB introduces DSConv. These DSConvs allow for an additional level of analysis, considering both channel and spatial dimensions, such as height and width, simultaneously. By incorporating DSConv, the DSEB aims to capture more intricate patterns and nuanced spatial information within the features. This enables the network to learn contextually rich representations that adapt better to the specific spatial characteristics of the data. The DSEB combines the strengths of the SE block, which focuses on channel-wise attention, with the added spatial adaptability brought by DSConv. This integration contributes to improved feature extraction, allowing neural networks to discern and emphasize important features across both channel and spatial dimensions more effectively. As shown in Figure 2, the main components in DSEB are as follows.

Depthwise convolution (DConv): This block leverages depthwise convolutions, which perform convolutions independently for each channel of the input, preserving spatial information while processing each channel separately. By doing so, it adapts to spatial characteristics, enabling better feature extraction [42]. Pointwise convolution (PConv): Following the DConv, a PConv is utilized. This PConv involves using 1 × 1 kernels to merge details across various channels through a linear combination derived from the DConv output. Its primary function is to expand the feature map’s depth while maintaining the spatial dimensions or potentially decreasing them. Frequently termed as the ‘bottleneck’ layer, the PConv reduces channel count, aiding in diminishing computational complexity [42]. Squeeze phase: Similar to the SE block, the DSEB begins with a “squeeze” phase. It globally pools features across spatial dimensions, generating channel-wise statistics to capture the importance of each channel’s information [43]. Excitation phase: Following the squeeze phase, learned channel-wise statistics are utilized to generate attention weights. These weights represent the relevance or significance of each channel in contributing to the final representation. This process enables the network to focus more on informative channels while suppressing less relevant ones [43]. Scale: The computed attention weights are applied to the original feature map to emphasize important channel-wise information. This scaled information is then fused with the output of the depthwise convolutional layer, enriching the feature representation with both spatial and channel-wise contextual information [43].

The DSEB’s innovation lies in its ability to combine the advantages of SE blocks’ channel-wise attention with the adaptability of DSConvs to capture spatial contexts efficiently. This integration results in improved feature representations, aiding the network in learning more discriminative and relevant features, which can lead to better performance in various computer vision tasks, such as semantic segmentation, object detection, and image classification.

### 3.2. Modified Inception Module (MIM)

In the Inception module (IM), convolution and maximum pooling operations are executed in parallel to extract richer features. DSConv divides the standard convolution into DConv and PConv. DConv performs independent convolutions for each channel of the input image, while PConv is used to combine the output obtained from DConv. This approach reduces the computational cost and the number of parameters by treating the standard convolution as two separate convolutions. By combining DSConv with the IM, a module with higher success rates and fewer parameters has been designed, termed as the MIM. This modified module, illustrated in Figure 3, replaces the sequential 1 × 1 − 3 × 3 convolution layers in the standard IM with 3 × 3 DConv and 1 × 1 PConv and replaces the 1 × 1 − 5 × 5 convolution layers with 5 × 5 DConv and 1 × 1 PConv [44]. The parameter counts for the standard and modified IMs are listed in Table 6 and Table 7. While the standard IM has a total parameter count of 15.488, the MIM has 1132 parameters. This adjustment significantly decreases the number of parameters used in computations, approximately by 13 times, resulting in highly successful outcomes.

### 3.3. ConvMixer Architecture

The ConvMixer (CM) presents a straightforward convolutional design suggested as an option to the patch-based structure of the Vision Transformer (ViT). While ViT achieves excellent results with self-attention layers, their computational time escalates quadratically and necessitates patch embeddings. Conversely, the CM works directly with input patches and relies solely on conventional convolutions for mixing steps. It ensures consistent resolution and uniformity throughout the network while independently addressing channel and spatial dimension integration [41]. The CM architecture outperforms both classical vision models such as ResNets and some corresponding MLP-Mixer and ViT variants, even with additions intended to make those architectures more performant on smaller datasets. The method is based on the idea of mixing, where DConv is used to mix spatial locations and PConv to mix channel locations. The method is instantiated with four hyperparameters: the hidden dimension, depth, kernel size, and patch size. The architecture is named after its hidden dimension and depth, like CM-h/d. The CM supports variable-sized inputs and is based on the idea of mixing, which is used in other architectures. These results suggest that patch embeddings themselves may be a critical component of newer architectures like ViT [41]. CM architecture consists of three parts, as shown in Figure 4. The first part of this architecture consists of a patch embedding layer and repeated applications of a fully convolutional block. The initial stage of this architecture incorporates a patch embedding layer, followed by multiple iterations of a fully convolutional block. Patch embeddings are applied through convolution, with specified input channels, output channels, kernel size, and stride. This process transforms an n × n image into a feature map of dimensions h × n/p × n/p, where p × p represents the patch size and h denotes the number of filters in the convolution layers the number of filters used in the convolution layer [45]. After the patch embedding layer, the GELU activation function is applied, succeeded by BatchNorm layers. Unlike RELU, GELU adjusts the inputs based on their magnitude instead of their sign, making it an efficient activation function. The second stage consists of the CM block, which is repeated a defined number of times based on the architecture’s depth. Each CM block includes a DConv operation, followed by a PConv operation, with every convolution being accompanied by an activation and BatchNorm step. The DConv operation is implemented within a residual block, which combines the input of a previous layer with the output of the subsequent layer to create a unified structure. DConv processes input channels independently, facilitating the mixing of spatial information within the image. PConv, on the other hand, employs a 1×1 convolution to analyze each pixel or point individually, allowing for the integration of data across patches.

After many applications of this block, GAP is performed to obtain a feature vector, which is passed to a softmax classifier [41]. This is the third part of the CM architecture.

## 4. Experimental Studies

### 4.1. AMD Datasets

In this study, two different datasets are used to determine the effectiveness of the proposed model. The first of these datasets is our private dataset obtained from the outpatient clinic of the Department of Ophthalmology at Elazığ Fethi Sekin City Hospital in Turkey. In total, 2316 OCT images from 653 eyes, collected from 256 participants, are included in the dataset. The dataset consists of 3 classes (dry AMD, normal, and wet AMD), with 653 images obtained from 89 patients for dry AMD, 743 images obtained from 82 patients for wet AMD, and 920 images obtained from 85 healthy individuals. Images from AMD samples within our dataset utilized for experimental analysis are depicted in Figure 5a. Dry AMD occurs when the macula thins and dries out, accompanied by the accumulation of a small quantity of shapeless material, known as drusen, within the eye’s cells. On the other hand, wet AMD involves an abnormal blood vessel developing beneath the macula, termed choroidal neovascularization. This vessel’s presence may elevate the macula due to leakage of fluid and blood, disrupting its flat position.

The second dataset used in the study is the public dataset. The Noor Eye Hospital in Tehran obtained this dataset through the use of imaging tools known as Heidelberg Spectral Domain Optical Coherence Tomography. Within this collection, there are 50 sets of normal OCT volumes, 48 sets related to dry AMD, and 50 sets connected to DME. Each of these sets contains a varied number of OCT images, ranging between 17 and 73 images. Altogether, these volumes encompass a sum of 1585 images for normal cases, 1637 for dry AMD cases, and 1104 for DME cases. The dimensions of the scans in this dataset measure 8.9 × 7.4 mm^2^, and the resolution along the axis is 3.5 µm. This particular dataset is also recognized by the name “Noor dataset”. Sample images of the Noor dataset are shown in Figure 5b.

### 4.2. Experimental Setup

The AMD dataset underwent experimental analysis using the TPU VM v3-8 hardware accelerator in the Kaggle platform. To train the proposed architecture, images sized at 224 × 224 × 3, with a batch size of 128 and a split of 15%–15%–70% for validation, testing, and training, respectively, were employed. In the dataset of 2316 AMD images, 347 of them are used for testing, 348 for validation, and 1621 for training. The same hyperparameters and training setup were also applied to experiments conducted on the public Noor dataset to ensure consistency across all evaluations. The model went through 100 training epochs, employing the Adam optimizer to minimize the loss function and enhance the model’s optimization. No data augmentation was applied during the study. Alongside hyperparameters, two specific callbacks played a crucial role in refining the training process. The initial callback, ReduceLROnPlateau, intervened by reducing the learning rate when validation loss stagnated, ensuring stability in training and countering overfitting. Within this callback, the minimum learning rate (lr) threshold was set at 0.000001, with a reduction factor of 0.3 for adjusting the lr. During training, the ModelCheckpoint, as the second callback, stored model weights periodically, ensuring the retention of the best model determined by validation accuracy, which can be utilized later on.

### 4.3. Evaluation Metrics

This research study employed various evaluation metrics, including F1 score (F1s), precision (Pr), accuracy (Acc), and recall (Re), to assess the classification performance of the PM. The Acc of the model is derived from Equation (1), where correct predictions are divided by the total predictions made. The Pr of the model measures its accuracy in positive predictions, as demonstrated in Equation (2). The model’s Re evaluates the accuracy of positive class predictions and is calculated using Equation (3). Additionally, the F1s, which represents a balanced measure of Pr and Re, is calculated following the steps outlined in Equation (4).
(1)$$\mathrm{Accuracy}\;(\mathrm{Acc})=\frac{\mathrm{True}\;\mathrm{Positives}+\mathrm{True}\;\mathrm{Negatives}}{\mathrm{True}\;\mathrm{Positives}+\mathrm{False}\;\mathrm{Positives}+\mathrm{True}\;\mathrm{Negatives}+\mathrm{False}\;\mathrm{Negatives}}$$


(2)
$$\mathrm{Precision}\;(\mathrm{Pr})=\frac{\mathrm{True}\;\mathrm{Positives}}{\mathrm{True}\;\mathrm{Positives}+\mathrm{False}\;\mathrm{Positives}}$$



(3)
Recall (Re)=True PositivesTrue Positives+False Negatives



(4)
$$F1\mathrm{score}\;(F1s)=2\;\times \frac{\mathrm{Precision}\;\times \;\mathrm{Recall}}{\mathrm{Precision}+\mathrm{Recall}}$$


The values of true negatives, false negatives, false positives, and true positives found in Equation (1)–(4) are extracted from the confusion matrix. These particular values are described below: True positives indicate the number of accurately classified input images within each class. False negatives correspond to instances where an image from a particular class is mistakenly classified as negative. True negatives represent correctly classified images that do not belong to that class. False positives indicate the count of incorrectly classified images within a specific class.

### 4.4. Experimental Results

The private AMD dataset used in experimental studies contains 3 classes: dry AMD, normal, and wet AMD. The confusion matrix obtained using test images in this AMD dataset is shown in Figure 6. The test dataset comprises 15% of the total number of samples. That is, out of a total of 2316 AMD images, 347 were used for testing. Upon examining the confusion matrix in Figure 6, the results indicate that out of a total of 104 images in the dry AMD class, 98 were correctly classified, 1 image was misclassified as normal, and 5 images were misclassified as wet AMD. All 133 images in the normal class were correctly classified, while out of 110 images in the wet AMD class, 109 were accurately classified. Only 1 of the wet AMD images was mistakenly classified as dry AMD.

The public AMD dataset (Noor dataset) used in experimental studies contains 3 classes: AMD, normal, and DME. The confusion matrix obtained using the test images in this AMD dataset is shown in Figure 7. The test dataset constitutes 15% of the total sample count. So, out of a total of 4326 AMD images, 649 were used for testing. Upon examining the confusion matrix in Figure 7, it is observed that all 248 AMD, 243 normal, and 158 DME images were correctly classified. Accordingly, Acc, Pr, Re, and F1s values for all classes were obtained as 100% based on this information.

Using our private dataset, the PM has been compared with different DL models, and the classification results are provided in Table 8. The DL models used for comparison are as follows: VGG16 [46], ResNet50 [47], ResNet101 [47], InceptionV3 [48], DenseNet121 [49], DenseNet201 [49], EfficientNetB0 [50], EfficientNetB7 [50], MobileNet [51], and ConvNeXtTiny [52]. When examining Table 8, the Acc, Pr, Re, and F1s values obtained by the PM are as follows: 97.98%, 97.95%, 97.77%, and 97.86%, respectively. The closest values to the PM are found in the DenseNet201 model, with 95.97% Acc, 95.74% Pr, 95.51% Re, and 95.62% F1s. The PM outperforms the DenseNet201 model by 2.01% in Acc, 2.21% in Pr, 2.26% in Re, and 2.24% in F1s. Similarly, compared to InceptionV3, the PM exhibits better performance by 2.3% in Acc, 2.46% in Pr, 3.24% in Re, and 2.85% in F1s. Additionally, in comparison to DenseNet121, the PM achieves superior classification performance by 2.3% in Acc, 2.77% in Pr, 2.34% in Re, and 2.56% in F1s. The classification results obtained from comparing other models to the PM are as follows: When compared to VGG16, the PM exhibits better performance by 4.32% in Acc, 4.79% in Pr, 4.74% in Re, and 4.77% in F1s. In comparison to ResNet50, the PM shows better performance by 6.91% in Acc, 7.52% in Pr, 7.88% in Re, and 7.7% in F1s. When compared to ResNet101, the PM demonstrates better performance by 7.2% in Acc, 7.8% in Pr, 8.1% in Re, and 7.95% in F1s. The experimental studies resulted in the following Acc, Pr, Re, and F1s values for models with the lowest classification performance: ConvNeXtTiny achieved 59.37% Acc, 65.16% Pr, 56.35% Re, and 60.44% F1s. EfficientNetB0 yielded 77.23% Acc, 76.16% Pr, 75.03% Re, and 75.59% F1s. MobileNet showed 79.25% Acc, 78.10% Pr, 77.63% Re, and 77.86% F1s. Lastly, EfficientNetB7 achieved 84.15% Acc, 82.27% Pr, 81.48% Re, and 81.87% F1s. When comparing all models, the results suggest that the classification outcomes acquired with the PM are highly successful. Additionally, Table 8 provides the parameter counts for all models. Upon examining the parameter counts in Table 8, it can be inferred that the PM achieves superior results with the least number of parameters.

In experimental studies conducted using the public (Noor) dataset, the PM has been compared with various DL models such as VGG16, ResNet50, ResNet101, InceptionV3, DenseNet121, DenseNet201, EfficientNetB0, EfficientNetB7, MobileNet, and ConvNeXtTiny. The results are presented in Table 9. Upon examining Table 9, it is observed that the PM achieved a value of 100% in all evaluation metrics. Among other methods, the results closest to the PM were obtained with the DenseNet121, achieving 99.85% Acc and 99.86% Pr, Re, and F1s. Following that, DenseNet201 achieved 99.69% Acc, 99.72% Pr, 99.65% Re, and 99.68% F1s. Additionally, InceptionV3 attained 99.23% Acc, 99.26% Pr, 99.31% Re, and 99.28% F1s. While ResNet50 yielded 98.92% Acc, 99.06% Pr, 98.82% Re, and 98.94% F1s, ResNet101 achieved 98.92% Acc, 98.98% Pr, 98.97% Re, and 98.97% F1s. VGG16 recorded 97.69% Acc, 97.70% Pr, 97.81% Re, and 97.75% F1s. EfficientNetB0 showed 91.52% Acc, 91.17% Pr, 91.48% Re, and 91.32% F1s, while EfficientNetB7 demonstrated 90.29% Acc, 90.61% Pr, 89.56% Re, and 90.08% F1s. Finally, MobileNet achieved 91.22% Acc, 90.73% Pr, 91.13% Re, and 90.93% F1s, whereas ConvNeXtTiny resulted in 62.86% Acc, 68.65% Pr, 59.84% Re, and 63.94% F1s. Upon examining all methods, the findings suggest that the PM achieved more successful results compared to the methods used for comparison. Furthermore, considering the trainable parameters of all methods, it is observed that the PM has the least number of trainable parameters. Considering all results, the PM has achieved significantly successful results with fewer parameters.

Additionally, training–validation (accuracy and loss) convergence curves for the PM using private and public (Noor) datasets are given in Figure 8. Figure 8a shows the training–validation curve for the private dataset, while Figure 8b shows the training–validation curve for the public (Noor) dataset. When examining Figure 8a, it is observed that the training accuracy increases up to approximately 20 epochs, then converges towards 100% after 20 epochs. The validation accuracy remains around 40% until approximately 60 epochs, after which it starts to increase. Considering the loss values, the training loss decreases up to about 20 epochs and converges to 0 after 20 epochs. The validation loss shows some fluctuations until around 60 epochs and converges to a value close to 0 after 60 epochs. Similarly, when examining Figure 8b, the training accuracy increases up to 20 epochs and reaches 100% after 20 epochs. The training loss also decreases up to 20 epochs and becomes 0 after 20 epochs. The validation accuracy shows fluctuations at certain intervals up to approximately 40 epochs and increases at certain intervals, reaching 100% after 40 epochs. The validation loss fluctuates until around 40 epochs and reaches 0 after 40 epochs. This indicates that both the training and validation accuracy values reach 100%. Considering the training–validation accuracy and loss curves for both private and public datasets, it is clear from the results that the training process converges rapidly. The rapid convergence of training–validation accuracy and loss curves in Figure 8b demonstrates the absence of overfitting, even with 100% accuracy achieved on the public dataset (Noor). The alignment of training and validation performance highlights the robustness and generalization capability of the proposed model.

### 4.5. Discussion

In this section, comparisons are made with existing methods in the literature using the public (Noor) dataset, and the results are given in Table 10. Paima et al. [25] introduced an innovative approach using a multi-scale CNN technique integrated with a feature pyramid network (FPN) for feature fusion. Their FPN-based method leverages diverse-scale receptive fields to enhance the detection of retinal abnormalities found at different sizes within OCT images. This approach enables seamless training of the multi-scale model through a singular CNN, employing a straightforward structure and obviating the necessity for preprocessing the input data. Within this methodology, the multi-scale CNN framework adopts preexisting architectures like VGG16, ResNet50, DenseNet121, and EfficientNetB0 as backbone structures. As a result of the experimental studies carried out using the Noor dataset, 87.80% Acc and 86.60% Re values were obtained when EfficientNetB0 was used with FPN. When DenseNet121 was used as the backbone structure with FPN, 90.9% Acc and 90.5% Re values were found. Similarly, when VGG16 was used as the backbone network with FPN, 92% Acc and 91.8% Re values were obtained, while when ResNet50 was used as the backbone network, 90.1% Acc and 89.8% Re values were obtained. When all backbone networks were examined, more successful results were obtained when VGG16 was used together with FPN. Thomas et al. [27] suggested an architectural design featuring a CNN that operates across multiple scales and paths, encompassing six layers of convolution. With the multi-scale convolutional layer, the network can generate diverse local structures employing filters of different sizes. Through multipath feature extraction, the CNN effectively amalgamates features associated with both sparse local and intricate global structures. The effectiveness of this architecture underwent evaluation via ten-fold cross-validation, employing various classifiers like support vector machines, multilayer perceptrons, and random forests. Experiments conducted on the Noor dataset showcased notable outcomes: using RF as a classifier alongside CNN achieved 98.97% Acc and 99% Pr, Re, and F1s. Employing SVM with CNN yielded 94.89% Acc and 94.9% Pr, Re, and F1s, while utilizing a multilayer perceptron classifier resulted in 96.93% Acc, 97% Pr, and 96.9% Re and F1s. Considering the comprehensive results, the most promising outcomes were achieved by combining CNN with the RF classifier. Das et al. [38] created a B-scan attentive CNN designed to replicate ophthalmologists’ diagnostic approach by concentrating on clinically significant B-scans while categorizing OCT images. To achieve this, they employed a feature extraction module based on CNNs, extracting spatial features from B-scans. Subsequently, a self-attention module gathered these features based on their clinical significance, resulting in a discriminative high-level feature vector for accurate diagnostics. Their trials with the Noor dataset demonstrated Acc of 94.9% and a Re rate of 93.26%. Rasti et al. [39] engineered an ensemble model that blends multi-scale convolutions for classifying DEM, dry AMD, and normal images sourced from OCT scans. Through empirical investigations on the Noor dataset encompassing these classes, their model achieved a Pr rate of 99.39%, Re rate of 99.36%, and F1s of 99.34%. Sabi et al. [53] introduce a new framework termed Double-Scale CNN designed for the diagnosis of AMD. This proposed structure integrates six convolutional layers to differentiate normal and AMD images. The double-scale convolutional layer enables the formation of multiple localized structures through the utilization of two distinct filter sizes. Within this proposed network, the sigmoid function serves as the classifier. Experimental investigations carried out on the Noor dataset unveiled an Acc level of 94.89%. Sahoo et al. [54] purposed to devise a prediction model for diagnosing dry AMD using a weighted majority voting (WMV) ensemble approach. This method merges forecasts from foundational classifiers and selects the predominant class based on weights assigned to each classifier’s prediction. A unique technique for extracting features along the retinal pigment epithelium (RPE) layer plays a crucial role in identifying dry AMD/normal images using the WMV method, wherein the number of windows calculated for each image is a determining factor. The preprocessing involves a hybrid-median filter, followed by scale-invariant feature transform-based segmentation of the RPE layer and curvature flattening of the retina, which aids in precisely measuring the RPE layer’s thickness. Through experimental analysis on the Noor dataset, they observed Acc of 96.94%, Pr of 95.83%, Re of 97.87%, and F1s of 96.84%. As per the lesion characteristics observed in OCT images, Xu et al. [55] introduced a hybrid attention approach, which combines a spatial attention mechanism (SAM) with a channel attention mechanism (CAM) operating concurrently. The CAM assigns distinct coefficients to individual channel feature maps within the channel dimension, aiming to emphasize the channel feature map containing the most prevalent lesion traits. Conversely, the SAM allocates specific position coefficients to each element in the spatial dimension, highlighting the significance of elements within the lesion area. This hybrid attention mechanism enables the network to prioritize the lesion area across both channel and spatial dimensions. Through experimental evaluation on the Noor dataset, they identified Acc of 99.76%, Pr of 99.70%, Re of 99.79%, and F1s of 99.74%. Mishra et al. [54] introduced an approach that incorporates an innovative deformation-responsive attention mechanism alongside pre-existing deep CNN architectures to derive improved distinguishing features from OCT images. This approach, labeled MacularNet, utilizes the VGG16 pre-trained model. Upon conducting experimental assessments on the Noor dataset to gauge the efficacy of the amalgamation of VGG16 and the attention mechanism, they found Acc of 99.79%, Pr of 99.80%, and Re and F1s of 99.79%. Fang et al. [56] introduced an innovative CNN approach tailored for the classification of OCT images, termed the lesion-aware CNN. Inspired by the diagnostic process followed by ophthalmologists, they devised a lesion detection network capable of identifying diverse types of macular lesions and generating corresponding attention maps. These attention maps, upon detection, are employed to delicately adjust the convolutional feature maps of the classification network, allowing the lesion-aware CNN model to emphasize crucial information, particularly related to macular lesions. Informed by lesion-specific data, the classification network harnesses insights from localized lesion-related areas, enhancing the efficiency and precision of OCT classification. Additionally, VGG16 serves as the foundational network for constructing the lesion-aware CNN model. Notably, the lesion-attention module is integrated into VGG16 before each pooling layer to retain more distinctive features through the maximum pooling process. To test the effectiveness of the lesion-aware CNN model, experimental studies on the Noor dataset yielded 99.39% Pr, 99.33% Re, and 99.36% F1s. Finally, a hybrid method has been developed in the PM, consisting of a combination of CM architecture, DSEB, and the MIM. The primary objective in developing the hybrid method is to reduce the trainable parameter count, thereby increasing classification accuracy while offering computational efficiency. As a result of experimental studies on the Noor dataset, the PM achieved a classification result of 100% across all evaluation metrics.

Based on the above information, it is observed that the most successful results were obtained using the PM. Results closest to the PM were achieved with Mishra et al. [54]’s MacularNet. The PM outperformed MacularNet by 0.21% in Acc, 0.2% in Pr, 0.21% in Re, and 0.21% in F1s. Alongside these results, the primary aim of the PM is to reduce the number of trainable parameters. While the PM has 1.650.020 parameters, MacularNet has approximately 19 million parameters. The PM holds about 12 times fewer parameters than MacularNet. Additionally, compared to Xu et al. [55]’s multi-branch hybrid attention network, the PM demonstrates a 0.24% improvement in Acc, 0.3% in Pr, 0.21% in Re, and 0.26% in F1s. Similarly, the PM outperforms Fang et al. [56]’s lesion-aware CNN method by 0.61% in Pr, 0.67% in Re, and 0.64% in F1s. Furthermore, the lesion-aware CNN method utilizes VGG16 as its base network, significantly increasing computational costs. Another study that achieved results closest to the PM is the multipath CNN with six convolutional layers and RF classifier method developed by Thomas et al. [27]. The PM yielded results that were 1.03% higher in Acc, 1% in Pr, 1% in Re, and 1% in F1s compared to this method. Additionally, the PM has about 4 times fewer parameters than the method developed by Thomas et al. [27]. The inclusion of the Noor dataset, originating from Tehran, Iran, demonstrates that the proposed model can generalize well to data from different geographic regions and imaging systems. The excellent performance of the PM on this dataset (100% across all metrics) highlights its robustness. However, future work will focus on expanding the dataset further to include images from additional geographic regions and diverse OCT devices to enhance the model’s applicability. The PM achieved a 5.11% higher Acc than the Double-Scale CNN method presented by Sabi et al. [53]. Moreover, compared to the multi-scale convolutional mixture of expert ensemble model suggested by Rasti et al. [39], the PM produced results that were 0.61% higher Pr, 0.64% higher Re, and 0.66% higher F1s. When contrasted with the B-scan attentive CNN method proposed by Das et al. [38], the PM showed a 5.1% higher Acc. Lastly, when compared to the FPN-VGG16 by Paima et al. [25], the PM outperformed it by 8%, FPN-ResNet50 by 9.9%, FPN-DenseNet121 by 9.1%, and FPN-EfficientNetB0 by 12.2% in Acc. It is important to note that the VGG16, ResNet50, DenseNet121, and EfficientNetB0 models used in the proposed methods by Paima et al. [25] significantly increase computational costs.

### 4.6. Ablation Study

The PM integrates a fusion of the CM, DSEB, and MIM design. This model comprises three distinct components: MIM, DSEB, and CM. The examination of individual components within PM was conducted with both private and public (Noor) datasets and is detailed in Table 11. For the private dataset, in the first 3 models, the results obtained from the individual use of the models are listed. In Model 1, only MIM was used, resulting in an Acc of 61.67%, Pr of 58.34%, Re of 57.80%, and F1s of 58.07%. In Model 2, only DSEB was used, yielding Acc, Pr, Re, and F1s values of 63.69%, 61.24%, 59.35%, and 60.28%, respectively. Model 3 exclusively utilized CM, resulting in an Acc of 95.10%, Pr of 95.16%, Re of 94.68%, and an F1s of 94.92%. In these three models, the findings suggest that the best result when used individually was achieved with CM. In Model 4, CM was used alongside DSEB. The incorporation of DSEB has positively influenced all evaluation metrics compared to using CM alone. When CM and DSEB were used together, it increased Acc by 0.58%, Pr by 0.39%, Re by 0.68%, and the F1s by 0.53% compared to CM alone. Similarly, in Model 5, MIM and CM were used together. Compared to using CM alone, it increased Acc by 1.44%, Pr by 0.79%, Re by 1.83%, and the F1s by 1.31%. In Model 6, MIM and DSEB were used together, resulting in Acc, Pr, Re, and F1s values of 66.28%, 61.37%, 61.62%, and 61.49%, respectively. The final model (Model 7) is the proposed one, utilizing all three components together. This combined model achieved an Acc of 97.98%, Pr of 97.95%, Re of 97.77%, and an F1s of 97.86%. Considering all models, the results indicate that the PM, composed of the combination of all three models, yields the best results.

When analyzing the components in the PM for the public (Noor) dataset, using all three components together (Model 7) resulted in 100% across all evaluation metrics. In Model 1, using only MIM yielded 82.13% Acc, 82.03% Pr, 82.84% Re, and 82.46% F1s. Model 2, using only DSEB, achieved 82.28% Acc, 81.61% Pr, 82.19% Re, and 81.90% F1s. Model 3, utilizing only CM, showed 99.54% Acc, 99.55% Pr, 99.50% Re, and 99.52% F1s. Individual usage reveals CM as the most successful model. In Model 4, combining DSEB and CM resulted in 99.69% Acc, 99.71% Pr, 99.59% Re, and 99.65% F1s. Model 5, combining MIM and CM, yielded 99.85% Acc, 99.86% Pr, 99.79% Re, and 99.82% F1s, while combining MIM and DSEB resulted in 92.91% Acc, 92.82% Pr, 92.90% Re, and 92.86% F1s. Upon examining all results, the findings indicate that the PM, which utilizes all three components together, performs successfully across the board.

## 5. Conclusions and Future Works

AMD, a progressive retinal condition, predominantly affects individuals aged 50 and above, leading to degenerative changes in the macula—a critical area of the retina responsible for central vision. It can lead to central vision loss, complicating everyday activities like reading and facial recognition. Advancements in ophthalmic imaging, particularly through technologies like OCT, have revolutionized the management and diagnosis of AMD by providing high-resolution, detailed insights into the retinal layers, enabling precise diagnosis and timely interventions. With the increasing prevalence of AMD, the demand for analyzing retinal scans is expected to grow, emphasizing the need for efficient diagnostic systems. This necessitates the development of automated diagnostic systems leveraging artificial intelligence (AI) to handle the growing number of OCT images efficiently. While the private dataset used in this study demonstrated high accuracy (97.98%), its relatively smaller size compared to the public dataset poses a potential limitation in terms of generalization. Increasing the training dataset size, especially through data collection from diverse regions, could further enhance the robustness and applicability of the proposed model. In addition to dataset size and diversity, it is important to acknowledge that clinical OCT scans often include noisy, artifact-laden, or incomplete images. These real-world challenges were not addressed in this study, as it primarily focused on high-quality datasets to establish baseline performance. Future research will aim to incorporate such challenging data to better evaluate the model’s robustness and reliability in real-world clinical scenarios. This step is crucial for ensuring the broader applicability of the proposed model in diverse clinical settings. The emergence of DL, particularly CNNs, has shown promise in significantly enhancing AMD detection and classification. The proposed CNN-based deep learning model in this study incorporates innovative architectural elements like CM architecture, DSEB, and the MIM. These modifications aim to improve computational efficiency, reduce parameter counts, and enhance feature extraction, leading to superior classification accuracy. The PM achieved an Acc of 97.98%, Pr of 97.95%, Re of 97.77%, and an F1s of 97.86% based on experimental studies conducted on a dataset consisting of 2316 images. In addition, as a result of the experimental studies carried out on the public (Noor) AMD dataset, 100% results were obtained in all evaluation metrics. Considering the experimental results, the findings suggest that the PM outperforms various existing DL models by attaining high Acc, Pr, Re, and F1s. These findings underscore the potential of AI-based systems, especially DL algorithms, in revolutionizing AMD diagnosis and treatment, providing a pathway for more accurate and efficient clinical practices in ophthalmology. In future works, the plan is to transform the PM into a real-time operational system applicable for expert doctors. Additionally, there are plans to extend the application of the PM to other eye diseases based on OCT images. While this study focused on three primary classes (normal, dry AMD, and wet AMD), we recognize that both dry AMD and wet AMD have additional progression stages and subtypes, such as geographic atrophy or variations in CNV presentation. Future research will aim to analyze these finer distinctions by incorporating detailed annotations for AMD subtypes and employing advanced visualization techniques, such as Grad-CAM, to better understand how the model differentiates overlapping features in these subtypes.

## Figures and Tables

**Figure 1 diagnostics-14-02836-f001:**
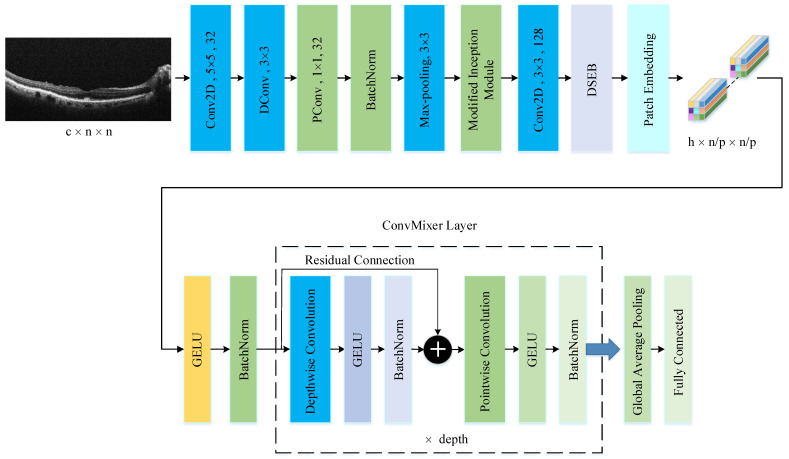
The structure of PM.

**Figure 2 diagnostics-14-02836-f002:**
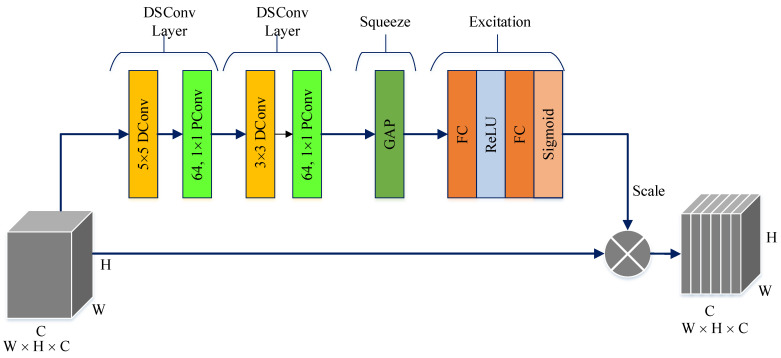
The structure of DSEB.

**Figure 3 diagnostics-14-02836-f003:**
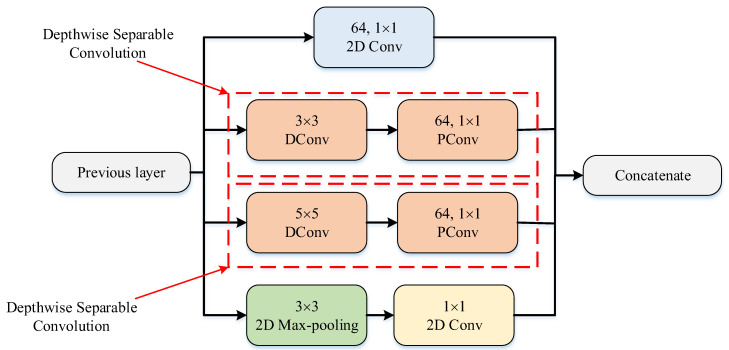
The structure of MIM.

**Figure 4 diagnostics-14-02836-f004:**
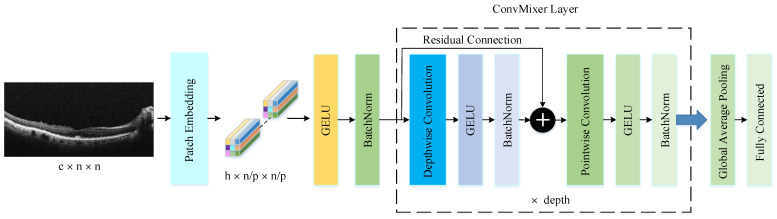
The structure of CM architecture.

**Figure 5 diagnostics-14-02836-f005:**
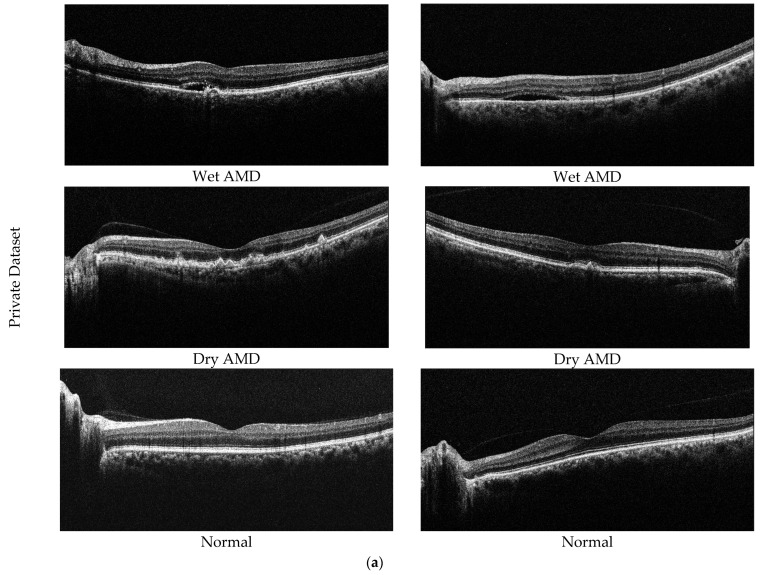
Sample AMD images in the datasets. (**a**) Private dataset and (**b**) Noor dataset.

**Figure 6 diagnostics-14-02836-f006:**
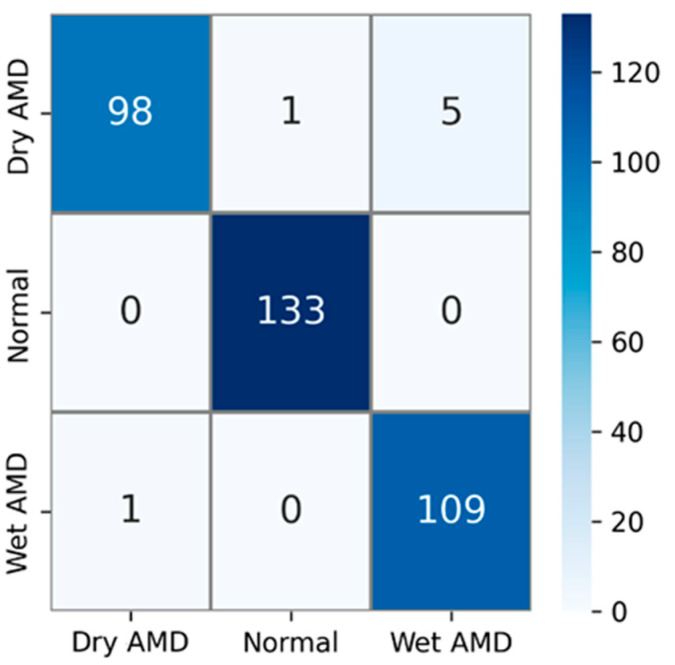
Confusion matrix for the PM using private dataset.

**Figure 7 diagnostics-14-02836-f007:**
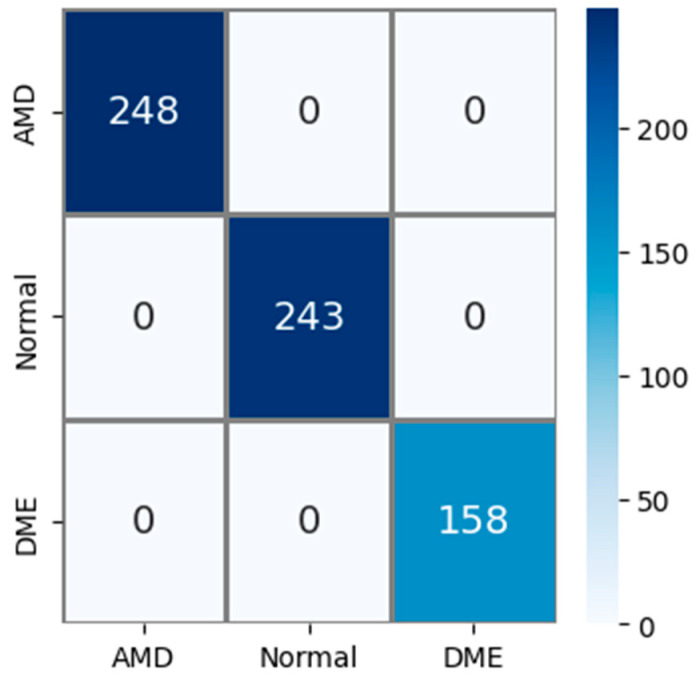
Confusion matrix for the PM using public (Noor) dataset.

**Figure 8 diagnostics-14-02836-f008:**
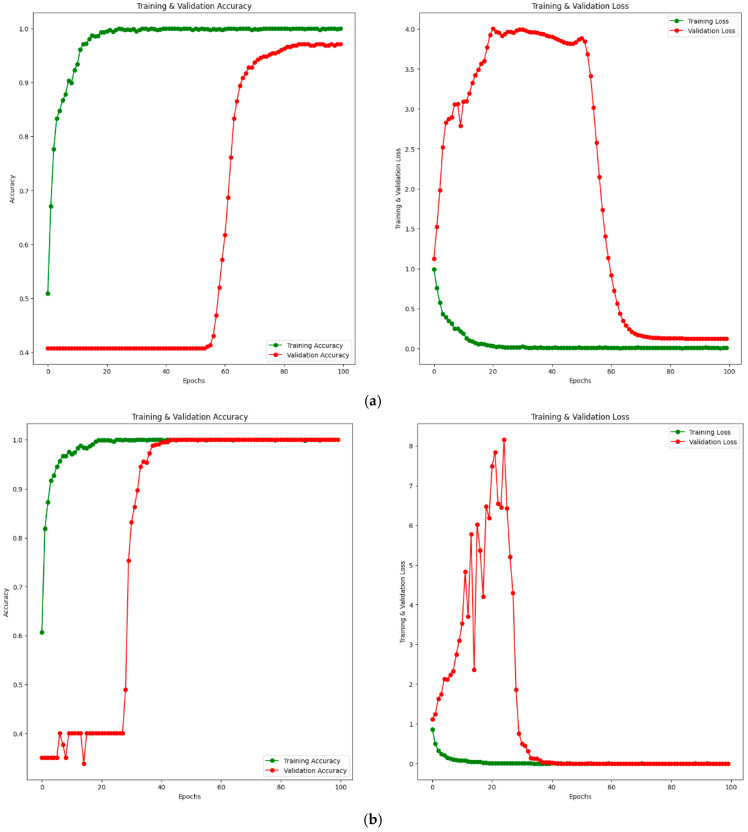
Training–validation convergence curves for PM. Accuracy curve on the left and loss curve on the right. (**a**) Private dataset. (**b**) Public (Noor) dataset.

**Table 1 diagnostics-14-02836-t001:** An overall overview of the research conducted in the literature employing DL-driven techniques for classifying AMD.

Authors	Datasets	Models	Results	Notes
Thomas et al. [21]	2130 images, 30 cases (2 classes: 15 dry AMD and 15 normal)	The speckle elimination relies on an adaptive denoising technique using contrast enhancement. To identify the retinal pigment epithelium layer, a method involving pixel grouping and iterative elimination is employed, utilizing knowledge about typical layer intensities and positions.	Acc: 96.66%	The primary focus of this study revolves around AMD classification employing a statistical method. By estimating drusen height through the retinal pigment epithelium layer and baseline, the severity of the disease becomes comprehensible. A step-by-step process, involving despeckling, retinal pigment epithelium and baseline estimation, drusen height detection, and classification, is implemented. Nevertheless, its efficacy diminishes when dealing with images where the majority of noisy pixels in columns exceed the intensity of the retinal pigment epithelium and are positioned below it.
Serener et al. [24]	UCSD [26] (Training: 26,315 normal, 8616 dry AMD, 11,348 DME, and 37,205 wet AMD Testing: 250 normal, 250 DME, 250 wet AMD, and 250 dry AMD)	Two pre-trained CNNs, ResNet18 and AlexNet, were used to classify OCT images associated with wet and dry AMD.	Dry AMD: ResNet18 (AUC: 94%, Acc: 99.5%) Dry AMD: AlexNet (AUC: 81%, Acc: 93.8%) Wet AMD: ResNet18 (AUC: 63%, Acc: 98.8%) Wet AMD: AlexNet (AUC: 61%, Acc: 96.5%)	In both instances (wet AMD and dry AMD), the ResNet18 outperformed the AlexNet. In addition, using pre-trained AlexNet and ResNet18 resulted in a large number of learnable parameters.
Paima et al. [25]	UCSD dataset [26] and Noor Eye Hospital OCT dataset (5667 normal, 3742 drusen, and 3240 CNV)	Multi-scale CNN with a feature pyramid network-based feature fusion strategy	Noor Eye Hospital OCT dataset (87.8% Acc for FPN-EfficientNetB0, 90.9% Acc for FPN-DenseNet121, 90.1% Acc for FPN-ResNet50, and 92.0% Acc for FPN-VGG16) UCSD dataset (93.9% Acc for FPN-VGG16, 5-fold cross-validation)	Within this research, the multi-scale CNN framework served as the backbone structure alongside CNN framework like VGG16, ResNet50, DenseNet121, and EfficientNetB0. Upon merging these architectures with the FPN framework, the most superior classification results were achieved using FPN-VGG16.

**Table 2 diagnostics-14-02836-t002:** An overall overview of the research conducted in the literature employing DL-driven techniques for classifying AMD.

Authors	Datasets	Models	Results	Notes
Kermany et al. [26]	UCSD dataset (4 classes: 51,140 normal, 8616 dry AMD, 37,205 wet AMD, and 11,349 DME)	An algorithm for transfer learning founded on the InceptionV3 model	Acc: 96.53%	This research showcased the effective capabilities of the transfer learning model, removing the necessity for an exceedingly specialized DL model and a dataset comprising millions of images. The classification process employed pre-trained InceptionV3 networks. However, their extensive number of learnable parameters and network complexity render them unsuitable for real-time applications.
Thomas et al. [27]	Dataset 1: (2 classes: 15 dry AMD and 15 normal) Dataset 2: (3 classes: 50 normal, 50 DME, and 48 dry AMD) Dataset 3: 2 classes: 115 normal and 269 AMD) Dataset 4: UCSD [26]	Multipath CNN with six convolutional layers and RF classifier	Acc = 96.66% for dataset 1, Acc = 98.97% for dataset 2, Acc = 99.74% for dataset 3, Acc = 99.78% for dataset 4 (Using OCT alone)	They introduced a CNN design that aims to aid AMD detection by analyzing OCT images. This CNN model consists of six convolutional layers using a multi-scale approach that enables the extraction of various local structures at various filter sizes. They achieved successful results using an RF classifier.
Celebi et al. [28]	Private dataset (266 normal, 156 wet AMD, 145 dry AMD, and 159 drusen), UCSD dataset	Capsule Network	Acc = 96.39% for private dataset, Acc = 98.07% for UCSD dataset (Using OCT alone)	The objective of the research is to enhance the precision of detecting early stages of AMD using a proposed Capsule Network trained on spectral domain OCT images with reduced speckle noise. This improvement is achieved through optimized augmentation techniques involving Bayesian non-local mean filters.
Hu et al. [29]	Private dataset 3401 OCT images (1167 drusen, 811 geographic atrophy, 711 nascent geographic atrophy, and 712 normal)	Hierarchical classification (EfficientNetV2, DenseNet169, Xception, and Normalizer-Free ResNet50)	The best F1S (92.08% Normalizer-Free ResNet50) Makro-F1s = 91.32% for four models (5-fold cross-validation)	This research introduces a hierarchical classification model. Within this model, the primary function of the base models is to extract features, which are subsequently employed to categorize images into their respective classes at varying hierarchical levels. Four CNNs—Normalizer-Free ResNet50, Xception, DenseNet169, and EfficientNetV2—were assessed as classification models for dry AMD. The most favorable outcomes were achieved using Normalizer-Free ResNet50 among these models.

**Table 3 diagnostics-14-02836-t003:** An overall overview of the research conducted in the literature employing DL-driven techniques for classifying AMD.

Authors	Datasets	Models	Results	Notes
Deng et al. [30]	Private dataset (Total 21 participants = 7 wet AMD, 7 normal, and 7 dry AMD, a total of 420 images, 20 images from each participant)	Texture features were extracted through the utilization of Gabor filters and non-linear energy transformation techniques, subsequently utilized to train a range of machine learning models, encompassing SVM, NN, and RF.	SVM: (normal Acc: 95.3%, dry AMD Acc: 93.1%, wet AMD Acc: 94.7%) NN: (normal Acc: 80.3%, dry AMD Acc: 73.1%, wet AMD Acc: 81.0%) RF: (normal Acc: 95.4%, dry AMD Acc: 80.0%, wet AMD Acc: 90.8%)	The proposed approach employs machine learning to identify AMD and differentiate between its various stages by utilizing choroidal images acquired through OCT. Texture characteristics are derived through Gabor filter banks and non-linear energy transformation. Subsequently, feature descriptors based on histograms are applied to train RF, SVM, and NN. These models were then evaluated on a dataset of choroid OCT images, featuring 21 participants.
Lee et al. [31]	Private OCT images dataset (48,312 AMD and 52,690 normal)	Modified VGG16	(Image Level) AUC = 92.78%, Acc = 87.63% (Macula Level) AUC = 93.83%, Acc = 88.98% (Patient Level) AUC = 97.45%, Acc = 93.45%	Employing a modified VGG16 model, the researchers attained significant classification accuracy. They acquired an area under the ROC curve of 92.78% for image-level analysis, 93.83% for macula-level assessment, and 97.45% for patient-level distinction, effectively discerning between normal and AMD images.
Yoo et al. [32]	83 cases (3 classes: 18 dry AMD, 38 wet AMD, 27 normal)	Feature extraction: VGG19 Classifier: RF	AUC = 90.6%, Acc = 82.6% (Using OCT alone)	They suggested VGG19 as a feature extractor, pre-trained on ImageNet photos, and a multiclass RF classifier to identify AMD images. On a limited dataset that included both OCT and matched fundus pictures, the total accuracy using OCT alone was 82.60%.
Hwang et al. [33]	Private dataset: 200 cases (4 classes: 968 dry AMD, 968 inactive wet AMD, 968 active wet AMD, 968 normal)	Pre-trained VGG16, ResNet50, InceptionV3	Acc = 90.73%(ResNet50) Acc = 92.67% (Inception V3) Acc = 91.40% (VGG16)	Among the three pre-trained models, InceptionV3 was found to be more successful.

**Table 4 diagnostics-14-02836-t004:** An overall overview of the research conducted in the literature employing DL-driven techniques for classifying AMD.

Authors	Datasets	Models	Results	Notes
Xu et al. [34]	OCT images dataset (821 cases (4 classes: 367 wet AMD, 62 dry AMD, 195 normal, 197 PCV)) Fundus images dataset (1099 cases (4 classes: 496 wet AMD, 107 dry AMD, 195 normal, 301 PCV))	ResNet50 and RF	Acc = 87.4%, Se = 88.8%, Sp = 95.6% (Using fundus and OCT)	They evaluated the effectiveness of a bi-modality deep CNN architecture in classifying AMD and PCV using OCT images and color fundus images. Each image was pre-labeled as PCV, dry or wet AMD, or normal. ResNet50 models served as the base, and alternative machine learning models such as RF classifiers were established for comparative analysis.
Chen et al. [35]	37,138 OCT images from 775 cases (2 classes: with or without lesions)	Ensemble model (AlexNet, VGG16, Inception V3, ResNet50, DenseNet)	Ensemble model Acc = 98.5%, Se = 98.7%, Sp = 98.4%, F1S = 97.7% (Using OCT alone)	They applied ensemble learning to screen retinal diseases and detect lesions using OCT images. The ensemble learning achieved better results than individual models.
Kadry et al. [36]	Private dataset (800 OCT and 800 fundus retinal images) (2 classes: AMD and non-AMD)	The handcrafted features, like the discrete wavelet transform, pyramid histogram of oriented gradients, and local binary pattern, are taken out from the test images and combined with VGG16’s DL features.	OCT: Acc: 97.50% and fundus retinal images: Acc: 97.08%	VGG16 was used to classify retinal images into non-AMD/AMD classes by combining deep and handcrafted features. In addition, basic handcrafted features were extracted from retinal test images through the discrete wavelet transform, local binary pattern methods, and pyramid histogram of oriented gradients which were combined with deep features to improve performance. The proposed combined deep and handcrafted feature technique using different binary classifiers achieved >97% accuracy for OCT and FRI images.
Kadry et al. [36]	Private dataset (800 OCT and 800 fundus retinal images) (2 classes: AMD and non-AMD)	Pre-trained VGG19, ResNet50, AlexNet, VGG16	Acc: VGG16: 86.66%, VGG19: 85.20%, AlexNet: 86.45% and ResNet50: 86.25%	They performed classification on 800 OCT and 800 fundus images from the Challenge AMD database and the OCT Image Database covering two classes: AMD and non-AMD. They used VGG19, ResNet50, AlexNet, and VGG16 for this task. The best accuracy was found with VGG16.

**Table 5 diagnostics-14-02836-t005:** An overall overview of the research conducted in the literature employing DL-driven techniques for classifying AMD.

Authors	Datasets	Models	Results	Notes
Das et al. [37]	UCSD dataset (4 classes: 51,140 normal, 8616 dry AMD, 37,205 wet AMD, and 11,349 DME)	Multi-scale deep feature fusion framework, aimed at enhancing classification efficiency	Acc: 99.6% Se: 99.6% Sp: 99.87%	Combining features from multiple scales enables the encompassment of variations spanning these scales, supplying supplementary details to the classifier. This approach eliminates the need for manually adjusting parameters to enhance accuracy. Nevertheless, the utilization of multiple CNNs amplifies the time needed for inference and the computational intricacy. Furthermore, to counter the class imbalance within the datasets, the loss function applied class-weighted categorical cross-entropy consistently during the learning phase.
Das et al. [38]	Noor Eye Hospital [39] and Duke [40] dataset	B-scan attentive CNN	94.9% Acc, 95% AUC for Noor Eye Hospital and 97.12% Acc, 97% AUC for Duke	An innovative B-scan attentive CNN is presented, mirroring the diagnostic approach of ophthalmologists. This method concentrates on clinically relevant B-scans for classification. Initially, a CNN-based feature extraction module is employed to derive spatial feature representations from the B-scans. Subsequently, a personal attention module combines these features based on their clinical significance to acquire a distinctive high-level feature vector, ensuring a dependable diagnosis.
Rasti et al. [39]	Noor Eye Hospital [39] dataset	A new model based on a multi-scale convolutional mixture of expert ensemble model to identify DME, dry AMD, and normal.	F1S: 99.34%, Recall: 99.36%, Precision: 99.39%, AUC: 99.8%	The methodology’s mathematical model was integrated with a fresh cost function that incorporates an extra cross-correlation penalty term. Achieving the best accuracy relies on manually adjusting the loss function. The utilization of multiple CNNs raises both computational complexity and inference time.
Proposed Model	Private dataset (2316 OCT images: 3 classes (653 dry AMD, 920 normal, and 743 wet AMD)) Public dataset: Noor Eye Hospital	A combination of the modified Inception modules, Depthwise Squeeze-and-Excitation Blocks, and ConvMixer	Private dataset: (F1S: 97.86% and Acc: 97.98%) Public Noor dataset: (Acc: 100% and F1s: 100%)	This study suggested a new CNN-based DL method for AMD diagnosis using the complexity of AMD and OCT images. The proposed model combines modified Inception modules, Depthwise Squeeze-and-Excitation Blocks, and ConvMixer architecture, offering computational efficiency while improving classification accuracy.

DME: diabetic macular edema, RFs: random forests, NN: neural networks, SVMs: support vector machines, Acc: accuracy, AUC: area under the curve, PCV: polypoidal choroidal vasculopathy, Se: sensitivity, Sp: specificity, F1S: F1 score, CNV: choroidal neovascularization.

**Table 6 diagnostics-14-02836-t006:** Number of parameters of the IM.

Layer	Input	Kernel_size/Filter	Number of Parameters
Input_Image	224 × 224 × 3	-	-
Conv1	Input_Image	1 × 1/64	256
Conv2	Input_Image	3 × 3/64	1792
Conv3	Conv2	1 × 1/64	4160
Conv4	Input_Image	5 × 5/64	4864
Conv5	Conv4	1 × 1/64	4160
Max_pooling	Input_Image	3 × 3/64	0
Conv6	Max_pooling	1 × 1/64	256
Total number of parameters			15,488

**Table 7 diagnostics-14-02836-t007:** Number of parameters of the MIM.

Layer	Input	Kernel_size/Filter	Number of Parameters
Input_Image	224 × 224 × 3	-	-
Conv1	Input_Image	1 × 1/64	256
DConv1	Input_Image	3 × 3/64	30
PConv1	DConv1	1 × 1/64	256
DConv2	Input_Image	5 × 5/64	78
PConv2	DConv2	1 × 1/64	256
Max_pooling	Input_Image	3 × 3/64	0
Conv 6	Max_pooling	1 × 1/64	256
Total number of parameters			1132

**Table 8 diagnostics-14-02836-t008:** Comparison with different DL models (for private dataset).

Model	Acc (%)	Pr (%)	Re (%)	F1s (%)	Parameters
VGG16	93.66	93.16	93.03	93.09	14,718,275
ResNet50	91.07	90.43	89.89	90.16	23,602,051
ResNet101	90.78	90.15	89.67	89.91	42,672,515
InceptionV3	95.68	95.49	94.53	95.01	21,817,123
DenseNet121	95.68	95.18	95.43	95.30	7,044,675
DenseNet201	95.97	95.74	95.51	95.62	18,335,427
EfficientNetB0	77.23	76.16	75.03	75.59	4,058,534
EfficientNetB7	84.15	82.27	81.48	81.87	64,115,610
MobileNet	79.25	78.10	77.63	77.86	3,236,035
ConvNeXtTiny	59.37	65.16	56.35	60.44	27,825,507
PM	97.98 *	97.95 *	97.77 *	97.86 *	1,650,020 *

* indicates the best result.

**Table 9 diagnostics-14-02836-t009:** Comparison with different DL models (for public Noor dataset).

Model	Acc (%)	Pr (%)	Re (%)	F1s (%)	Parameters
VGG16	97.69	97.70	97.81	97.75	14,718,275
ResNet50	98.92	99.06	98.82	98.94	23,602,051
ResNet101	98.92	98.98	98.97	98.97	42,672,515
InceptionV3	99.23	99.26	99.31	99.28	21,817,123
DenseNet121	99.85	99.86	99.86	99.86	7,044,675
DenseNet201	99.69	99.72	99.65	99.68	18,335,427
EfficientNetB0	91.52	91.17	91.48	91.32	4,058,534
EfficientNetB7	90.29	90.61	89.56	90.08	64,115,610
MobileNet	91.22	90.73	91.13	90.93	3,236,035
ConvNeXtTiny	62.86	68.65	59.84	63.94	27,825,507
PM	100 *	100 *	100 *	100 *	1,650,020 *

* indicates the best result.

**Table 10 diagnostics-14-02836-t010:** Comparison of methods in the literature using the public Noor dataset.

Study	Model	Acc (%)	Pr (%)	Re (%)	F1s (%)	Parameters
Paima et al. [25]	FPN and VGG16	92.00	-	91.80	-	-
	FPN and ResNet50	90.10	-	89.80		-
	FPN and DenseNet121	90.90	-	90.50	-	-
	FPN and EfficientNetB0	87.80	-	86.60	-	-
Thomas et al. [27]	Multipath CNN with six convolutional layers and RF classifier	98.97	99.00	99.00	99.00	6,024,512
Thomas et al. [27]	Multipath CNN with six convolutional layers and SVM classifier	94.89	94.90	94.90	94.90	6,024,512
Thomas et al. [27]	Multipath CNN with six convolutional layers and multilayer perceptron classifier	96.93	97.00	96.90	96.90	6,024,512
Das et al. [38]	B-scan attentive CNN	94.9	-	93.26	-	-
Rasti et al. [39]	Multi-scale convolutional mixture of expert ensemble model	-	99.39	99.36	99.34	-
Sabi et al. [53]	Double-Scale CNN	94.89	-	-	-	-
Sahoo et al. [57]	Adaptive window-based feature extraction and weighted ensemble-based classification approach	96.94	95.83	97.87	96.84	-
Xu et al. [55]	Multi-branch hybrid attention network	99.76	99.70	99.79	99.74	-
Mishra et al. [54]	MacularNet	99.79	99.80	99.79	99.79	Approximately 19 million
Fang et al. [56]	Lesion-aware CNN	-	99.39	99.33	99.36	-
PM	Combination of the modified Inception modules, Depthwise Squeeze-and-Excitation Blocks, and ConvMixer architecture	100 *	100 *	100 *	100 *	1,650,020 *

* indicates the best result.

**Table 11 diagnostics-14-02836-t011:** Analysis of components in the PM.

Model	MIM	DSEB	CM	Private Dataset				Public (Noor) Dataset			
				Acc (%)	Pr (%)	Re (%)	F1s (%)	Acc (%)	Pr (%)	Re (%)	F1s (%)
Model 1	X	-	-	61.67	58.34	57.80	58.07	82.13	82.08	82.84	82.46
Model 2	-	X	-	63.69	61.24	59.35	60.28	82.28	81.61	82.19	81.90
Model 3	-	-	X	95.10	95.16	94.68	94.92	99.54	99.55	99.50	99.52
Model 4	-	X	X	95.68	95.55	95.36	95.45	99.69	99.71	99.59	99.65
Model 5	X	-	X	96.54	95.95	96.51	96.23	99.85	99.86	99.79	99.82
Model 6	X	X	-	66.28	61.37	61.62	61.49	92.91	92.82	92.90	92.86
Model 7	X	X	X	97.98 *	97.95 *	97.77 *	97.86 *	100 *	100 *	100 *	100 *

* indicates the best result.

## Data Availability

Data will be made available on request.

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
