# Peer review of "A Comprehensive CNN Model for Age-Related Macular Degeneration Classification Using OCT: Integrating Inception Modules, SE Blocks, and ConvMixer"

_diagnostics, 2024, doi:10.3390/diagnostics14242836_

Round 1
Reviewer 1 Report
Comments and Suggestions for Authors
1) The paper significantly uses two datasets – a private and a public one. The former consists of a total of 2316 OCT images and the latter comprises of 4326 OCT images. By applying a standard 70%-15%-15% classification, the number of images used for training from dataset 1 and 2 is 1621 and 3028 respectively. Pretrained models that are compared here visualizes and extract features which indicates that the number of data is sufficient but increase in number of training data might provide accurate results.
2) In addition to the number of data, the dataset provided here sticks to a single geographic region i.e., Turkey. The real-world OCT images have several characteristic progression, age related variability due to genetic and environmental factors which the model should learn. The design and data acquisition may vary between the OCT devices in Hospitals. A model may fail if the input fed into it has been taken from another OCT device. This might be achieved by augmenting data from different geographic regions.
3) The proposed model has achieved 100% accuracy while using the public dataset. This might be due to the fact that the model overly recognizes the features in the dataset. The authors can check for model overfitting.
4) External clinical validation performances may be checked to get the proper validation loss of the model
Reviewer 2 Report
Comments and Suggestions for Authors
I read this article with interest, but I want to disclose that I am commenting on this article from the perspective of an ophthalmologist, not as an expert in computational systems. As an ophthalmologist reading the article, it becomes apparent that most of the authors are not experts in ophthalmology.
Abstract:
1) The tone of the abstract seems overly optimistic. Please introduce nuances to reflect the limitations and validity of your findings more accurately.
Introduction (seems more like it was written by AI than by a human):
2) The initial sentence, "The retina in the human eye plays a crucial role in interpreting visual data," sounds strange. Does the "retina" actually interpret visual data? Interpretation is primarily a function of the brain.
3) The sentence "Nerve tissues sensitive to light envelop the inner part of the eyeball" reads as though it belongs in a children’s book. It oversimplifies retinal anatomy.
4) "Directed by the lens, light reaches the retina, stimulating the creation of neural signals" is misleading. It is not only the lens that directs light; the cornea provides the majority of the eye's refractive power.
5) The statement "Situated within the retina, the macula..." is incorrect. The macula is not "situated within" the retina; it is a specific area of the retina.
6) "Macula ... and decoding visual specifics, hues, and brightness levels" is incorrect. The macula does not "decode" visual data. It captures signals and transmits them to the brain for interpretation.
7) Describing AMD as "an eye-related condition" is overly simplistic. Maybe retinal condition?
8) The phrase "AMD is categorized into two forms" is simplistic. Maybe use the verb "could"
9) "In those with dry AMD, the presence of drusen in the retinal pigment epithelium layer signifies a significant anomaly". Drusen are also present in wet AMD.
10) The phrase "This drusen buildup leads to a reduction and drying out of the macula, causing a loss of its function" is scientifically inaccurate. Avoid using "drying out of the macula," as it has no clear meaning.
11) "While individuals with dry AMD might retain sufficient central vision" could be misleading. This is true only in early stages of dry AMD, not necessarily in all cases.
12) "The key cause of wet AMD is choroidal neovascularization (CNV)" is inaccurate. CNV is one of the main causes but not the "key" cause.
13) "Eye specialists" --> "ophthalmologists"
14) Replace "AMD-related lesions" --> "AMD-related features"
15) "OCT serves to diagnose and track various eye conditions in a noninvasive manner" --> "various retinal conditions."
16) "Evaluating visual disorders such as optic nerve disease" --> "optic disc abnormalities."
17) "The shortcoming of fundus photography is that it only provides two-dimensional retinal information" it is not 100% true. Most OCT images are also 2D. The distinction lies in the resolution, sectional imaging, magnification, and the fact that fundus photography provides only a frontal view.
18) This a very ChatGPT-style paragraph "Through precise imaging, OCT allows for swift interventions, customized treatment strategies, and better patient results, enabling early detection. Its remarkable abilities have revolutionized the realm of eye care, empowering ophthalmologists to deliver more accurate diagnoses and tailored care for individuals grappling with retinal issues". Please avoid this type of genetic paragraphs. I marked in red all the typical ChatGPT expressions.
19) The statement "Nevertheless, as the prevalence of AMD increases, the frequency of eye examinations will rise, leading to busier practices and a reduced timeframe for ophthalmologists to analyze patient information" is debatable. This is not necessarily a real concern. Perhaps it would be more accurate to say that machines can assist in identifying subtle features better than humans.
20) Similarly, the claim "Consequently, as retinal scans become more intricate, there will be a rapid surge in the quantity of OCT images to be processed" lacks evidence. Again "rapid surge" is a very AI way to talk.
21) The phrase "Hence, there's an urgent requirement" overstates the situation. The term "urgent" is not justified here.
22) The expression "AI-based detection techniques can satisfy this desire"... Avoid using "satisfy this desire" in a technical context.
Main text
My main concerns related to the model presented by the authors
24) What about the private dataset? Was it validated? Is this dataset well balanced in terms of demographics, device variations, and disease severity?
25) The Noor dataset is only composed by 50 Normal, 48 AMD & 50 DME eyes. And it is very easy to distinguish the various form. So a very good performance here could be expected. Testing the PM on a third, diverse dataset could be valuable in order to evaluate its robustness in real-world clinical settings.
26) Is the model applicable in real-world? The study does not address the model's performance on low-quality or noisy images, a common issue in clinical OCT scans. Evaluating the PM on noisy, artifact-laden, or incomplete images would demonstrate its reliability in real-world scenarios.
27) Although the PM is compared to other DL models, the conditions for these comparisons are unclear. Were all models trained and evaluated on identical datasets with optimized hyperparameters? The study should explicitly describe how competing models were trained and benchmarked to ensure a fair comparison.
28) The PM appears to perform well on binary (normal vs. AMD) and three-class classification tasks (normal, dry AMD, and wet AMD). However, additional analysis is needed to assess performance across disease subtypes, particularly those with overlapping features.
Conclusions
Here again I see a very ChatGPT style
29) "AMD, a progressive eye ailment, predominantly affects individuals aged 50 and above, causing harm to the macula". AILMENT? HARM TO THE MACULA?
30) "The described advancements in ophthalmic imaging, particularly through technologies like OCT, have revolutionized the manage- ment and diagnosis of AMD. These imaging modalities, especially OCT". You are repeating the OCT concept twice.
31) "However, as the population affected by AMD increases, the volume of imaging data from retinal scans escalates, posing chal- lenges in rapid analysis and timely treatment." I do not agree about the "retinal scans escalates".
32) I Do not see any admission of shortcoming the conclusions. Authors should clearly be less optimistic (avoid sentences like "it's evident that the PM outperforms"... and highlight the problems in their study.
I leave to the other reviewers the specific comments about the computational models.
Comments on the Quality of English LanguagePlease avoid all the ChatGPT style in all the manuscript. This is too evident.
Round 2
Reviewer 2 Report
Comments and Suggestions for Authors
Thank you for revising your manuscript and adding more nuances to your results. However the manuscript still needs some corrections:
-
Some of your corrections have made the introduction a bit redundant and repetitive. Please strike out entirely, for example, the first sentence of the introduction: "The retina in the human eye captures visual information and transmits it to the brain for interpretation," and also: "Light enters the eye and is primarily focused by the cornea, with additional focusing by the lens, before reaching the retina where neural signals are generated."
-
As I suggested by me, you corrected "AMD could be categorized," but upon further reflection, it might be better to say "AMD may be categorized."
-
"Choroidal neovascularization (CNV), [...] is one of the primary pathological features of wet AMD" → change as "primary causes."
-
Remove "and patient outcomes" from line 101.
-
I recommend moving the text from lines 687–693 to the very end of the conclusions.
-
In my previous report, I suggested removing overly optimistic phrases like "It's evident that the PM outperforms." However, the specific sentence I mentioned has not been changed (I don't understand why). Please avoid saying "it's evident" and use more cautious language instead. This applies to lines 634, 644, 655, and—of course—695.
- Checking at the plagiarism from line 300 to 326, there are to many similarities with a previously published article from some of the same authors. I'm referring in particular to this article (paragraph "ConvMixer"): Demirbaş, A.A., Üzen, H. & Fırat, H. Spatial-attention ConvMixer architecture for classification and detection of gastrointestinal diseases using the Kvasir dataset. Health Inf Sci Syst 12, 32 (2024). https://doi.org/10.1007/s13755-024-00290-x
Thank you and congratulations for your hard work.
Author Response
1) Specifically, we have removed the first sentence, "The retina in the human eye captures visual information and transmits it to the brain for interpretation," and the sentence, "Light enters the eye and is primarily focused by the cornea, with additional focusing by the lens, before reaching the retina where neural signals are generated."
2) it has changed "may be categorized".
marked in red. located at line 54.
3) The updated sentence, located at lines 65-66, now reads:
"Choroidal neovascularization (CNV), which develops beneath the macula and retina, is one of the primary causes of wet AMD."
the modification has been marked in red.
4) Removed as suggested.
5) Moved as recommended.
6) We sincerely apologize for overlooking this issue in the previous revision. We appreciate your patience and have now addressed the concern by replacing "it's evident" with more cautious language, such as "the findings suggest that" or "the results indicate that," in lines 401, 441, 480, 630, 640 and 685. These changes are highlighted in the revised manuscript for your review.
7) Thank you for your observation. The section in lines 307 to 324 has been revised to reduce similarities with the referenced article. The updated version maintains the technical details while rephrasing and restructuring the content to ensure originality.